# Response Surface Method for Optimization of Synchronous Reluctance Motor Rotor

Svetlana Orlova [1],* , Janis Auzins [2], Vladislav Pugachov [1], Anton Rassõlkin [3] and Toomas Vaimann [3]

1 Laboratory of Modelling of Electromagnetic Processes, Institute of Physical Energetics, LV-1006 Riga, Latvia
2 Machine and Mechanism Dynamics Research Laboratory, Riga Technical University, LV-1048 Riga, Latvia
3 Department of Electrical Power Engineering and Mechatronics, Tallinn University of Technology, 12616 Tallinn, Estonia
* Correspondence: sorlova@edi.lv

**Abstract:** In order to define the best design structure of the synchronous reluctance motor (SynRM) rotor, optimization must be carried out, implying the selection of the best alternative for each specific criterion. The optimization of an electrical machine is a complicated work involving meeting different criteria requirements while dealing with a range of constraints. In order to implement the optimization, it is necessary to process a huge number of options, changing the combinations of the factors affecting criteria and restrictions, which is a time-consuming process. This research presents the optimization technique that gives a mathematically proven solution of the optimal rotor design of a synchronous reluctance machine obtained by using metamodels in the form of local polynomial approximations. Analysis of the results of numerical modeling and experimental investigation has been performed in order to validate the developed technique and recommendations. SynRM rotor was manufactured, with the stator to be taken from the 1.1 kW W21 WEG induction motor, which makes possible the relevant experimental study. The performance analysis of the developed SynRM is shown in the paper.

**Keywords:** synchronous reluctance motors; design optimization; design of experiments; response surface methodology

## 1. Introduction

This paper is an extended version of the authors' 2021 28th International Workshop on Electric Drives: Improving Reliability of Electric Drives paper [1] that was presented in January 2021 in Moscow, Russia.

More than 53% of the electricity is consumed worldwide by electric motor systems used in industry, buildings, structures, agriculture, and transport [2,3], generating approximately 6040 Mt of $CO_2$ emissions per year. Therefore, the European Union (EU), as well as the United States (US), China, and other countries have adopted legislation that makes it mandatory to apply gradually increasing energy efficiency requirements in relation to new equipment units. The International Electrotechnical Commission (IEC) has introduced energy efficiency classification for electric motors, also known as IE codes, as stipulated in the international standard IEC 60034-30-1 [4]. According to the international standard, there are currently four energy efficiency classes for electric motors, IE1, IE2, IE3, and IE4 (IE—International Energy Efficiency Class). Increasing the energy efficiency requirements encourages researchers to develop alternative technologies for electric machines. One of the options to reach IE3 and IE4 efficiency classes is using rare-earth permanent magnets in electric machines, ultimately causing a relatively larger impact on the environment. Currently, around eight billion electric motors are being used in the EU, consuming about half of the electricity produced in the EU. The sector is very variegated, with a considerable range of technologies, applications, and sizes, ranging from small motors (e.g., motors that run computer cooling fans) to huge motors used in the heavy industry. The directive

on the eco design of energy-using products is to be replaced by a regulation laying down eco-design requirements for electric motors and variable speed drives. The new legal framework will include not previously covered asynchronous motors (small motors ranging from 0.12 kW to 0.75 kW; large motors from 375 kW to 1000 kW) [5]. In addition, the requirements will be toughened as three-phase motors with a nominal power between 0.75 kW and 1000 kW or less have to reach IE3 class. Motors with a power range from 75 kW to 200 kW must comply with IE4 class requirements as of July 2023 [6]. The new rules will also regulate the efficiency of variable-speed motors, and both product groups will be subject to requirements such as efficiency at different load points, speed, and torque.

This will allow researchers to carry out general system optimization. In the same way as before, the motors designed for specific operational conditions will be excluded from the legal framework or subjected to more favorable rules.

An efficient motor can make savings from a few EUR up to several tens of thousands of EUR throughout its lifetime, depending on its power and operation mode [5]. Under the current legal framework, the most efficient motors in the EU provide energy savings of 57 TWh per year. In the light of the revised regulations, the savings will reach 110 TWh by 2030, which is equal to the annual electricity consumption in the whole country of the Netherlands. This means that each year, $CO_2$ emissions will be reduced by 40 million tons, and the annual energy bill of EU households and industries will decrease by around 20 billion EUR by 2030. Integrated optimization of the electric motor drive system (including the use of high-efficiency and well-designed components) is the main strategy for increasing the overall efficiency of electric motors [7,8].

Therefore, manufacturers are becoming more interested in synchronous reluctance motors [9,10]. In this research, the optimization procedure of SynRM aimed at improving torque, specific torque, and efficiency by response surface methodology is presented. Part of this research provides the optimization technique that gives a mathematically proven solution of the optimal rotor design of a SynRM. This work proposed a resource-saving technique for rotor shape optimization by applying metamodels through local polynomial approximation. Approximate mathematical models (metamodels) are often used as surrogates for more computationally intensive simulations. A number of researchers have developed suitable metamodels to reduce the computational time needed to solve complex structural problems.

## 2. Design Optimization

Optimization involves the best possible methods and procedures applied to find feasible solutions for different technical and mathematical objectives. It includes mathematical results and numerical methods aimed at finding and identifying the best possible alternatives from the variety of options. Optimization methods make it possible to choose the best option without direct testing and evaluation of a whole variety of available options. They are closely linked to the use of mathematical methods, logical procedures, and algorithms implemented by means of computer hardware.

In order to determine the best rotor design for a motor, an optimization procedure must be performed with the selection of the best option for each criterion. The electrical machine design optimization procedure is a complicated work associated with the observance of various criteria in the presence of a number of restrictions. The rational and optimum design of a magnetic system depends on different factors attributable to the operation of an electromagnetic device. Motor torque, efficiency coefficient, power coefficient, and all main characteristics of a machine are defined by the difference in magnetic conductivity in axes *d* and *q*.

Saliency ratio, defined as the ratio of the *d* axis inductance to the *q* axis inductance, is the most important parameter in designing a synchronous reluctance motor, which has a major impact on achieving both maximum power factor and maximum torque. The goal of optimization of synchronous reluctance motor is the large saliency ratio *Ld/Lq* value.

Despite the use of high-performance computers (HPC) and even multi-cluster computing centers for parallel calculations, addressing numerous actual technical problems, particularly regarding optimization, requires too large computing resources. Therefore, the complex mathematical models should be preferably replaced with fast-track metamodels that ensure that during the optimization procedures, the necessary results are obtained within a reasonable time. Experiment planning helps in raising awareness about the investigation object [10].

E. P. Box and K. B. Wilson laid the grounds for the experimental optimization method called the Response Surface Method (RSM). Surface methodology is a collection of mathematical and statistical techniques useful for the modeling and analysis of problem in which a response of interest is influenced by several variables and the objective is to optimize this response [11]. They quickly found that improving the process according to linear models is not effective for real processes having the interaction of input factors and multiple optimum combinations. Publications on RSM root back to the article by Box and Wilson [12–15]. This paper had a major impact on industrial applications for experimental planning and was a huge motivation factor for many studies in this area. In general, the application of RSM has three purposes:

1. Mapping the response surface (display) in a specific interest area. This gives the designer understanding of what is expected as a result of changes in the parameters of the system or process. For this purpose, different graphical displays are applied: 3D surfaces and counter plots. The problem is that in the case of a larger number of input factors, it is only possible to create section graphics, i.e., record some factors with constant values, and view the other two factors graphically.

2. Optimization of the response. Optimization using computer programs is usually not difficult for approximated models. If there is more than one optimization criterion, Pareto ideology or method weighted criteria can be applied. However, optimal result validation is always required. In addition, it may appear that the approximated model has a major error, and its optimum is not applicable to the physical model. Then the whole RSM process must be repeated, adding experimental tests and other regression functions.

3. Change product or process parameters to adjust to standard specifications or customer requirements. The main problem of the response surface problem is the number of responses that should be analyzed simultaneously. If a client has determined a certain concentration in its project, the designer must reach this level at a minimum cost.

4. Today, the Response Surface Methodology has evolved into a Metamodeling Methodology but is often also referred to as the Response Surface Method. The most significant development of RSM began with the onset of numerical experiments. There are mathematical models for numerical experiments often the Finite Elements (FE) model is applied. In this model, it is possible to calculate responses at the given input parameter values. However, the relationship between input factors and responses is not analytically describable. Numerical experiments and approximations are carried out to obtain an understandable mathematical model. These approximations are the exact approximation of the FE model, which is referred to as a metamodel or surrogate model [10].

As a part of this research, it provides the optimization technique that gives mathematically proven solution of the optimal rotor construction of a synchronous reluctance motor. The current optimization was done based on algorithm:

- Aim and objective;
- Factors and range;
- Plan of numerical experiment;
- Modelling of experiment;
- Synthesis of metamodel;
- Verify prediction.

### 3. Design of the Experiment

The object of investigation is a synchronous reluctance motor with transversally laminated anisotropy rotor, four poles, and a double-layer distributed winding. Figure 1 shows SynRM rotor with three magnetic barriers and a cut-out design, radial and tangential ribs. Thin ribs are left at punching, thus various rotor segments are connected to each other by these ribs. The high number of holes and air inside the rotor makes the rotor structure weaker. A smooth motor torque is achieved by choosing the correct number of flux guides and flux barriers against the ratio of the stator slot number to the number of pole pairs. The size of flux barriers affect the amount of magnetic flux in the rotor both in d and $q$ axis. Thin flux barriers reduce the reluctance in $d$ axis which makes the $Ld$ inductance grow, but at the same time $Lq$ inductance increases [16–19].

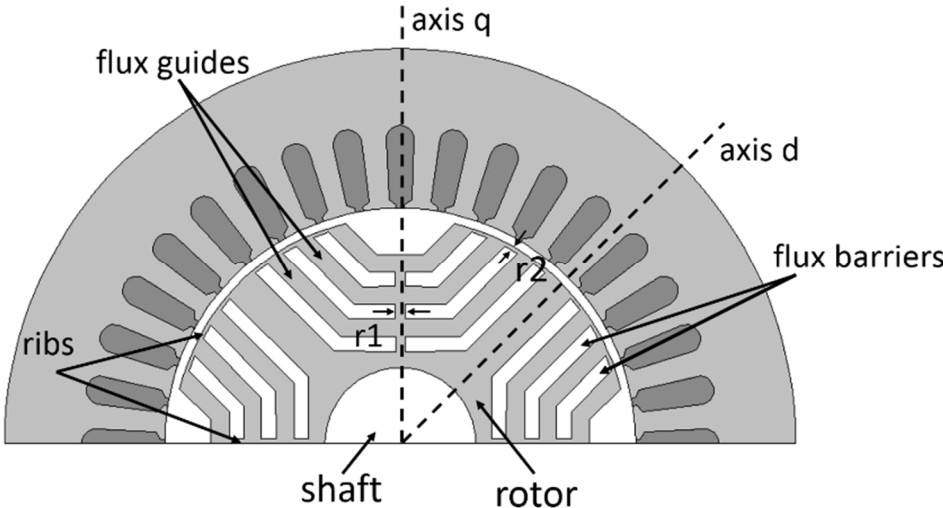

**Figure 1.** View of investigated synchronous reluctance motor ($r_1$-radial rib; $r_2$-tangential rib).

The main parameters to be chosen during the designing of the optimum rotor are as follows: the number of magnetic barriers, the air/iron ratio, the width of radial and tangential ribs, the air gap width, and the number of poles. A smooth motor torque is achieved by choosing the correct number of flux guides and flux barriers against the ratio of the stator slot number to the number of pole pairs.

The stator of SynRM is the same as the one of an induction motor and was taken from 1.1 kW motor with IE2 efficiency class. The stator parameters are presented in Table 1. Main technical data of the induction motor: 1.1 kW; $U$ = 380 V; $f$ = 50 Hz; $I$ = 2.69A; $cos \phi$ = 0.76; efficiency $\eta$ = 81.6%; $M$ = 7.22 Nm; $2p$ = 4, weight 18 kg.

**Table 1.** Main motor stator design parameters.

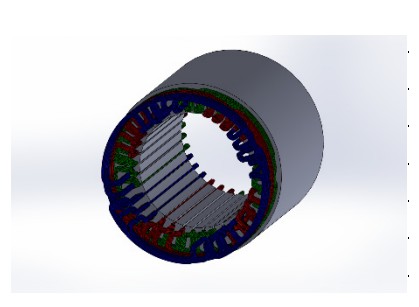

| Parameters | Value |
| --- | --- |
| Number of slots | 36 |
| Outer diameter | 139 mm |
| Inner diameter | 92.7 mm |
| Stack length | 110 mm |
| Number of turns | 47 |
| Number of parallel paths | 2 |
| Coil pitch | 1–8 |
| Filling factor | 69.48% |
| Wire diameter | 0.56 mm |

In this stator, short-pitched winding with a coil span from 1 to 8 is applied. The short-pitched winding allows to improve the wave-form of generated electromotive force, also eddy-current and hysteresis losses are reduced.

The saliency ratio of the studied SynRM is obtained by inserting flux barriers in the transversally laminated rotor, as shown in Figure 1. The number of flux barriers and their thickness determines how much magnetic flux can penetrate the rotor in *d*- and *q*-axes. Flux in the *d*-axis should be as high as possible, while flux in the *q*-axis should be minimized. To minimize the flux in the *q*-axis, the flux barrier should be as wide as possible, but at the same time, the amount of iron in the *d*-axis is reduced, which causes the *d*-axis flux to decrease. This is why it is so important to find the right thicknesses of flux barriers giving the maximized saliency ratio.

For a correct evaluation of the flux barrier width and flux guide width, the coefficient insulation ratio is used (1):

$$k_w = \frac{w_b}{w_g}, \tag{1}$$

where $w_b$ is the sum of the flux air barrier widths, and $w_g$ is the sum of the flux guide widths.

The coefficient $k_w = 0$ means that the rotor is made completely of iron (no saliency), while $k_w = 1$ signifies that the rotor is designed of lamination segments in which the numbers of air barriers and flux guides are equal. In [20–24], was shown that an optimum inductance is reached when this ratio is approximately 50:50 (i.e., $k_w = 1$). In order to check this statement, the insulation ratio range is chosen from 0.2 to 1.2 in Table 2.

**Table 2.** Maximum and minimum range.

|  | Variables | Unit | Limits |
|---|---|---|---|
| $x_1$ | Rotor outer radius | mm | $45.25 < R < 46.05$ |
| $x_2$ | Radial rib | mm | $1 < r_1 < 3$ |
| $x_3$ | Tangential rib | mm | $1 < r_2 < 3$ |
| $x_4$ | Insulation ratio | - | $0.2 < k_w < 1.2$ |
| $x_5$ | Number of barriers | - | $1 < b < 5$ |

It is defining the boundaries and factors that should be independent variables and will affect the result. For the synthesis of metamodels, it is necessary to choose the variable parameters and their ranges and to conduct various calculations of the researched machine using the method of FE. The chosen range of variable parameters is presented in Table 2.

In order to define the best design structure of the SynRM rotor, optimization was carried out, implying the selection of the best alternative to follow the specific criterion (Table 3).

**Table 3.** The objective of the optimization.

|  | Variables | Unit | Aim |
|---|---|---|---|
| $y_1$ | Torque | Nm | Maximize |
| $y_2$ | Specific torque | Nm/kg | Maximize |
| $y_3$ | Efficiency | % | Maximize |

In order to analyze and optimize synchronous reluctance motor with transversally laminated anisotropy (TLA) rotor, metamodels describing how the selected variable parameters affect motor torque, specific torque, and efficiency must be synthesized, considering the constraints. The limitation factors are magnetic induction values of the rotor, stator yoke, and stator tooth.

The relation between the response variable *y* and independent variables is unknown. In general, the low-order polynomial model is used to describe the response surface *f*.

The model with $N$ experimental runs is carrying out on $q$ design variables and response $y$ as follows:

$$y_i = \beta_0 + \beta_1 x_{i1} + \beta_2 x_{i2} + \ldots\ldots + \beta_q x_{iq} + \varepsilon_i \ (i = 1, 2, \ldots\ldots, N) \tag{2}$$

The response $y$ is a function, $f$, of the design variables $x_1, x_{2,\ldots,}x_q$, plus the experimental error and $\beta_i{}'$ are regression coefficients [25].

The Latin hypercube plan of various experiments consisting of 31 combinations of variable parameters for the TLA rotor has been drafted. For generating an experimental plan, EDAOpt software [10], developed by RTU Scientific research of machine and mechanism dynamics laboratory, was applied; experimental plan LH type is optimized using mean square error. The software EDAOpt provides all phases of experimental optimization: (1) design of experiments, (2) creating a mathematical model on the basis of experimental results, (3) multi-objective and robust optimization using approximated models as objective and constraint functions, and (4) validation of results.

The variable parameters, their ranges, and quantity, as well as the number of implementing experiments, may differ depending on the problem being solved to implement numerical experiments using specialized software for the calculation of the electromagnetic fields. To model the magnetic field of an electrical machine and to calculate the necessary physical values, there are a lot of electromagnetic field simulation softwares that can be used, such as AMPERES (three dimensional (3D)), VSim (3D), OERSTED (two dimensional (2D)), ANSYS Maxwell 30 (2D/3D), QuickField (2D), MagNet (2D/3D), etc. Numerical modeling of the SynRM electromagnetic field of a range of experiments was performed using finite-element software.

The experimental results were produced in accordance with the selected range (Table 2) and the experimental plan. The base of synthesis of the metamodel is the approximation of the experimental data. The metamodels were synthesized based on the results of numerical calculations of the magnetic field. The main advantages of metamodels are the ability to perform simulations and understand the interconnection of input and output parameters, as well as the prompt implementation of the optimization processes. During work implementation, the original plan was supplemented with additional points where validation was performed. The points giving the highest response $y_3$ values during modeling were entered as five equal points. Thus, they have given the greatest weight to the method of least squares. Thus, in the last stage, the number of points is 59, of which only 49 are different. Figure 2 is shown the optimum Legendre polynomial items number, which is selected based on cross-validation value. Cross-validation is one of the most widely used data resampling methods to estimate the true prediction error of models and to tune model parameters [21].

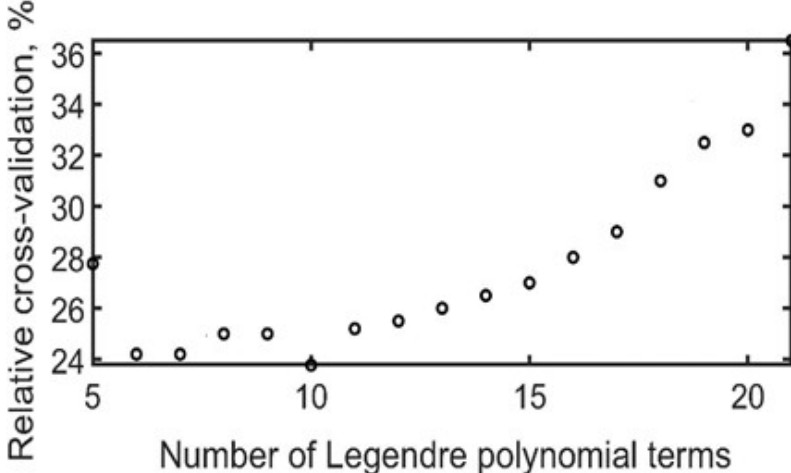

**Figure 2.** Cross-validation value vs a number of Legendre polynomial terms.

Cross-validation [22–24] is any of various similar model validation techniques for assessing how the results of a statistical analysis will generalize to an independent data set. Cross-validation is a resampling method that uses different portions of the data to test and train a model on different iterations. It is mainly used in settings where the goal is prediction, and one wants to estimate how accurately a predictive model will perform in practice The goal of cross-validation is to test the model's ability to predict new data that was not used in estimating it, in order to flag problems like overfitting or selection bias [24] and to give an insight on how the model will generalize to an independent dataset.

Cross-validation belongs to the family of Monte Carlo methods.

This metamodel consists of 10 items, giving a minimal cross-validation error—23.77. The approximation of $y_3$ in the Legendre polynomial notation is as follows (3):

$$\hat{y}_2 = 3.904699(0000) + 0.490127(10000) + 0.114892(00001) - 0.097149(00100) + 0.045091(01010) + 0.043736(20000) \\ -0.030252(10010) + 0.00603(01001) - 0.036858(01000) - 0.46118(00020) \tag{3}$$

The approximation error for $y_3$ is near $\pm 0.04$. $y_1$(torque), $y_2$(specific torque), and $y_3$ (efficiency) were approximated by a polynomial of order 2 to obtain the following expressions (4)–(6):

$$f_1 = 25253.277 - 1117047.5 \times x_1 + 26101.851 \times x_2 + 13149.888 \times x_3 + 47.35254 \times x_4 + 9.2066189 \times x_5 + \\ +12353099 \times x_1 \times x_1 - 574995.59 \times x_1 \times x_2 - 298694.35 \times x_1 \times x_3 - 945.50735 \times x_1 \times x_4 - 203.20315 \times x_1 \times x_5 \\ -2.697632 \times x_4 \times x_4 \tag{4}$$

$$f_2 = 7675.4708 - 338918.76 \times x_1 + 12031.648 \times x_2 + 3938.5846 \times x_3 - 27.226954 \times x_4 + 2.1118292 \times x_5 + \\ +3741112.6 \times x_1 \times x_1 - 264025.42 \times x_1 \times x_2 - 83449.317 \times x_1 \times x_3 + 648.37478 \times x_1 \times x_4 - 45.904875 \times x_1 \times x_5 - \\ -387.98711 \times x_3 \times x_4 - 0.74784492 \times x_4 \times x_4 \tag{5}$$

$$f_3 = 347.2761 - 15495.836 \times x_1 - 52.5603078 \times x_2 - 1113.5657 \times x_3 + 4.7053763 \times x_4 - 1.0203362 \times x_5 + \\ +173181.56 \times x_1 \times x_1 - 1989.5639 \times x_1 \times x_2 + 24625.091 \times x_1 \times x_3 - 101.48301 \times x_1 \times x_4 + 22.371051 \times x_1 \times x_5 + \\ +12292.393 \times x_2 \times x_2 + 12565.293 \times x_2 \times x_3 + 51.802368 \times x_2 \times x_4 + 7.3741892 \times x_2 \times x_5 - \\ -15966.402 \times x_3 \times x_3 - 0.11350708 \times x_4 \times x_4 \tag{6}$$

Residuals are the difference between registered and approximated values of the response at experimental points (7):

$$\text{Res}_{m,i} = y_{m,i} - \hat{y}_m(x_i), i = 1, 2, \ldots, N \tag{7}$$

where $m$—number of the response, $i$—number of the experimental run.

In creating an approximation model, it is important that the residuals are as small as possible in absolute terms—this is ensured by the least square method. It is also important that the distribution of the responses is close to the normal distribution and that the dependence of the residuals on the response is not recorded. Residuals distribution is presented on Figure 3.

As can be seen, $y_3$ residuals have little dependence on registered values—the regression line is not precisely horizontal. But this dependence is small and tolerable.

As Appendix Figure A1 shows, the histograms of the residuals are nearly similar to the normal distribution. The normal probability plot was also tested in Figure A2. Graph points are not far from the straight line.

Unconstrained maximization of responses in Table 4.

Using metamodel and EdaOpt software, the optimal design of the SynRM rotor in five parameter differences was obtained. As can be seen from Figure 4, the rotor outer radius has a significant influence on the torque.

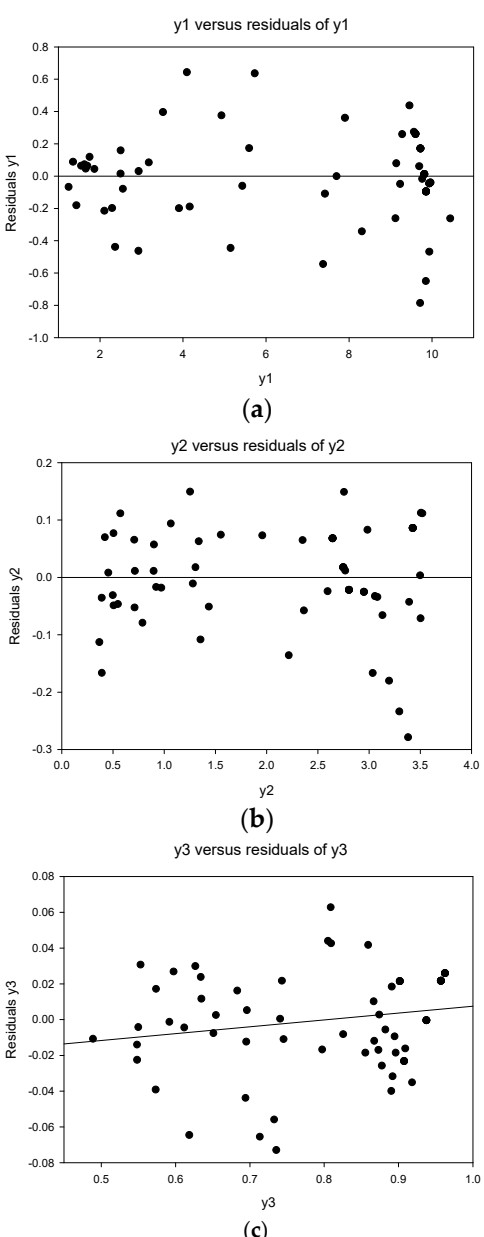

**Figure 3.** Residuals distribution of (**a**) $y_1$; (**b**) $y_2$; (**c**) predicted $y_3$.

**Table 4.** Response of EDAOpt software.

|  | $y_1$ | $y_2$ | $y_3$ |
|---|---|---|---|
| Criterion | −10.443548 | −3.5202343 | −0.95434034 |
| $x_1$ | 0.0461 | 0.0461 | 0.0461 |
| $x_2$ | 0.001 | 0.001 | 0.001 |
| $x_3$ | 0.001 | 0.001 | 0.001 |
| $x_4$ | 0.69776964 | 1.2 | 0.33468864 |
| $x_5$ | 1 | 1 | 5 |
| $y_1$ | 10.443548 | 9.7631095 | 9.4437398 |
| $y_2$ | 3.0903761 | 3.5202343 | 2.527117 |
| $y_3$ | 0.86956646 | 0.80096256 | 0.95687512 |

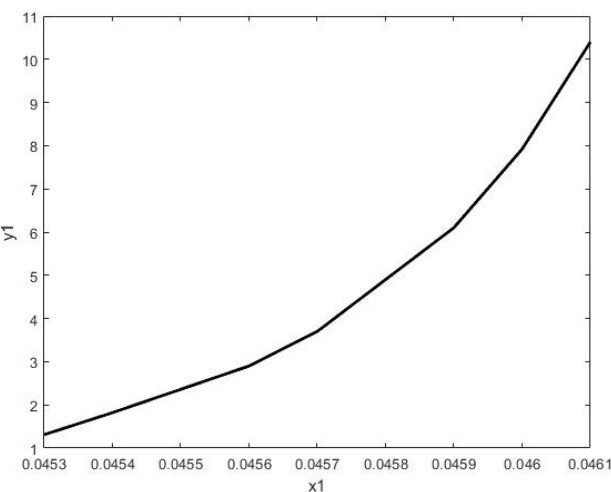

**Figure 4.** Torque ($y_1$) vs. rotor outer radius ($x_1$).

Pareto frontier is obtained using the Monte Carlo method by calculating the criteria values in a regular network with $600 \times 600 \times 600 \times 600 \times 5$ node points and selecting those points that are not competing. Of the 648 billion points, 3371 Pareto frontier points were selected, a 3D Mesh plot (Figure 5). The calculation takes about 15 h for the Intel i9 10900K 5.1 GHz processor.

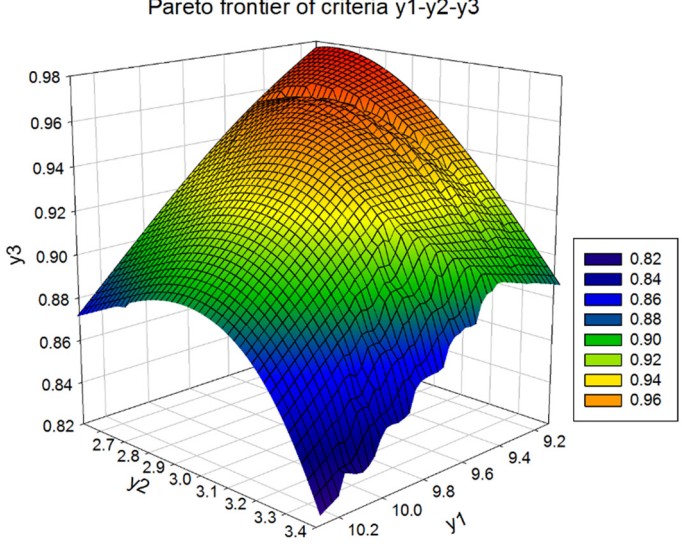

**Figure 5.** Contour plot of Pareto frontier of criteria $y_1, y_2, y_3$.

The energy efficiency criterion is opposite to the maximum torque as well as the maximum relative torque criterion. By maximizing torque, a motor with high efficiency will not be obtained. A small local efficiency increase zone is observed at $y_1 = 9.5$, $y_2 = 2.8$ (Figure 6). The input parameters are $x_1 = 0.046097$, $x_2 = 0.00114$, $x_3 = 0.00102$, $x_4 = 0.5295$, $x_5 = 5$. Using metamodel, the design of the SynRM rotor for the experimental model in five parameter differences was obtained. The optimal design of SynRM rotor: air gap height is equal 0.2 mm, radial rib—1 mm, tangential rib—1 mm, insulation ratio—1, number of barriers—4.

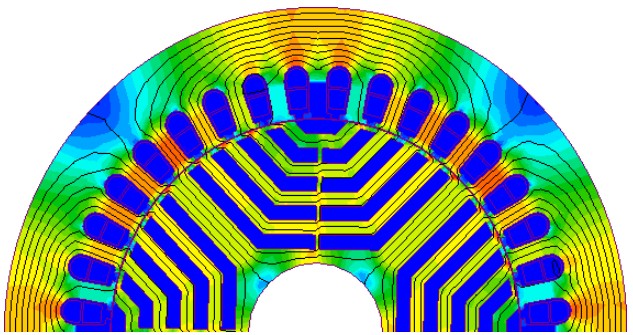

**Figure 6.** Magnetic flux and magnetic flux distribution in motor cross-section.

The metamodel optimization received data was approbated by a numerical experiment. Magnetic flux and flux density distribution of SynRM are presented in Figure 6, $x_d$ = 153 Ω, $x_q$ = 29.1 Ω.

## 4. Experimental Study

In the course of the experiments, the main energy characteristics of SynRM and IM were determined. The involved motors were powered by variable-frequency drive YASKAWA GA 500 or industrial applications. Motor control was implemented using open-loop vector control for SynRM and scalar control for IM.

To verify the proposed optimum rotor design, an experimental model was created; the model rotor and experimental setup are shown in Figures 7 and 8.

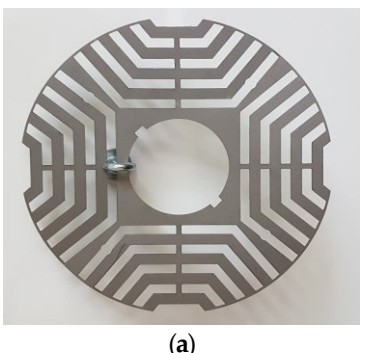
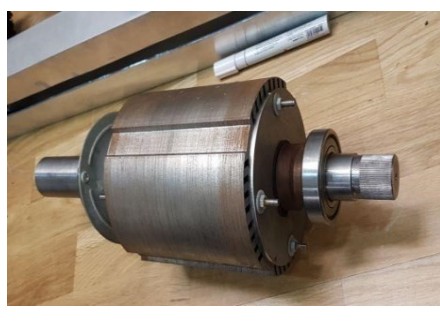

(**a**)  (**b**)

**Figure 7.** Cross-section of the rotor with transversally laminated anisotropy: (**a**) rotor sheet (**b**) experimental rotor model.

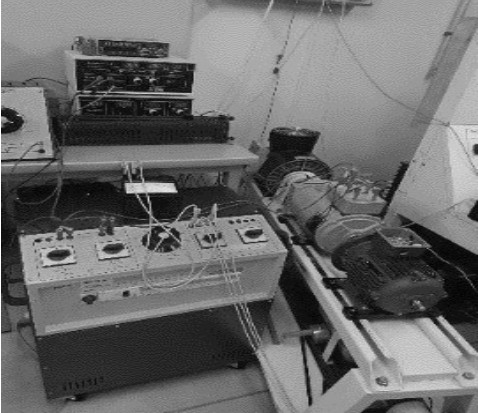

**Figure 8.** Experimental setup for motor testing.

The object of investigation is a synchronous reluctance motor based on a modified induction motor 1.1 kW WEG. Main technical data of the induction motor: 1.1 kW; $U = 380$ V; $f = 50$ Hz; $I = 2.69$ A; $cos\phi = 0.76$; efficiency $\eta = 81.6\%$; $M = 7.22$ Nm; $2p = 4$, weight 18 kg. Experimental research was carried out in the laboratory of Riga Technical University, with partial use of the equipment of the Institute of Physical Energetics.

In the course of the experiments, the main energy characteristics of SynRM were determined. The involved motors were actuated by the variable-frequency drive for industrial applications. Motor control was implemented using open-loop vector control. The no-load and load tests results are given in Tables 5 and 6, respectively. Current vs. torque and current vs. efficiency graphs of SynRM are given in Figure 9a, the maximum efficiency (89%) is achieved at torque equal to 7.5 Nm at a rated speed of 1500 rpm. A simulation test was performed to validate the proposed model that takes into account saturation, iron losses, and mechanical losses. Figure 9b shows the input power values as a function of the motor current for 1500 rpm.

**Table 5.** No-load test result of the SynRM.

| Voltage (V) | 220 | 290 | 340 | 385 |
|---|---|---|---|---|
| Phase current (A) | 1.33 | 1.33 | 1.33 | 1.33 |
| Speed (rpm) | 750 | 1050 | 1347 | 1500 |

**Table 6.** Load test result of the SynRM AT 1500 RPM.

| Voltage (V) | 380 | 380 | 380 | 380 | 380 | 380 |
|---|---|---|---|---|---|---|
| Phase current (A) | 1.6 | 2.08 | 2.33 | 2.7 | 3.49 | 5.01 |
| Input power (W) | 410 | 814 | 1118 | 1280 | 1650 | 2460 |
| Torque (Nm) | 2.5 | 4.5 | 6.5 | 7.5 | 8.5 | 11.5 |
| Output power (W) | 368 | 670 | 988 | 1140 | 1319 | 1770 |
| Efficiency | 87 | 87 | 88 | 89 | 80 | 72 |
| Speed (rpm) | 1500 | 1500 | 1500 | 1500 | 1500 | 1500 |

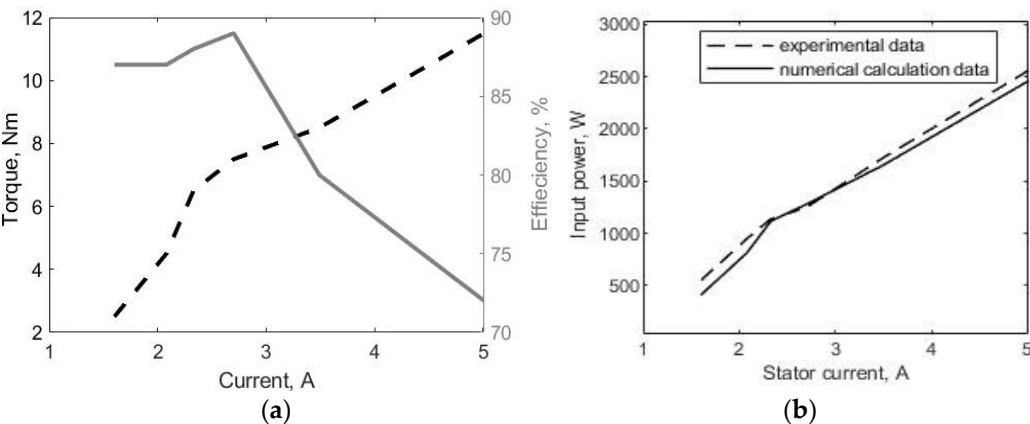

**Figure 9.** (**a**) Motor current vs. torque and efficiency and (**b**) SynRM input power values as a function of motor stator current for the case of 1500 rpm.

## 5. Conclusions

As part of this research, it presents the optimization technique that gives a mathematically proven solution for the optimal rotor construction of a synchronous reluctance machine with a TLA rotor. In this work, a resource-saving technique is proposed for shape optimization of the rotor by using metamodels in the form of local polynomial approximations. The technique is based on using computer-aided design software (CAD),

software based on the method of finite elements, and Matlab software. Legendre polynomials allow for building the metamodels with insignificant terms, thus increasing the accuracy of the prediction. It should be noted that the metamodel created with RSM allows multicriterial optimizations.

The received data of optimal motor rotor design were validated using numerical calculations and experimental work in the laboratory. Based on the comparison of the obtained data, it can be concluded that the developed technique could be used for general motor design. But it should be noted that energy savings of motors are also gained through the use of variable speed drive systems.

**Author Contributions:** Conceptualization, S.O. and V.P.; methodology, J.A.; software, S.O.; validation, S.O., V.P. and A.R.; resources, T.V. All authors have read and agreed to the published version of the manuscript.

**Funding:** This research is partly funded by the Latvian Council of Science, project "Creation of design of experiments and metamodeling methods for optimization of dynamics of multibody 3D systems interacting with bulk solids and fluids", project No. lzp-2018/2-0281. The research has been partly supported by the European Regional Development Fund within the project "Development of a high-efficiency rare-earth metal-free electric motor". No 1.1.1.2./VIAA/1/16/173.

**Conflicts of Interest:** The authors declare no conflict of interest.

## Appendix A

Figure A1 shows, the histograms of the residuals for three objectives of optimization are nearly similar to the normal distribution.

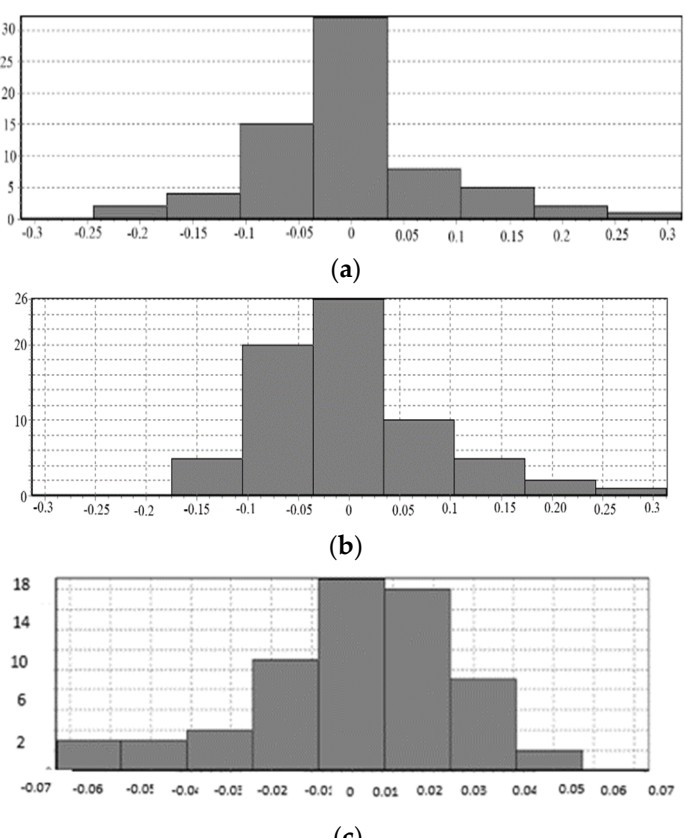

(a)

(b)

(c)

**Figure A1.** Histogram of residuals: (**a**) $y_1$; (**b**) $y_2$; (**c**) $y_3$.

The normal probability plot for motor efficiency ($y_3$) is shown on Figure A2.

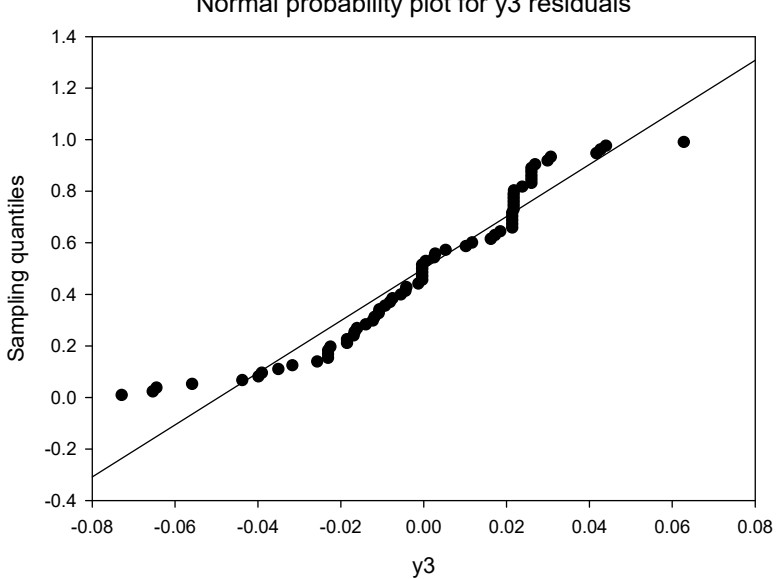

**Figure A2.** Normal probability plot for $y_3$ residuals.

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
