# Peer review of "Response Surface Method for Optimization of Synchronous Reluctance Motor Rotor"

_machines, doi:10.3390/machines10100897_

Round 1

Reviewer 1 Report

This paper proposed the optimization method fora mathematically proven solution of the optimal design of a synchronous reluctance machine using metamodels in the form of local polynomial approximations. The results of numerical modeling and experimental investigation has been performed.

Optimal designed method is key point in this paper; however, the statement of the theoretical analysis is difficult to be found in section 2. Moreover, the optimal goal for the designed motor should be given. The designed characteristics of the investigated motor shown in Fig. 1 should be explained. In section 3, the authors claimed an optimum inductance being reached when this ratio is approximately 50:50 (i.e. KW=1) and 0.2<KW<1.2 in Table 2, which should be explained. The formulation of the equations seems to be improved, as like (2)~(6). Too many figures and tables are given in section 3 and section 4, which should be clarified to fit the key point of the paper. For example, more statement should be added to clear show the idea of Figs. 4 and Fig. 9. More explanations to determine the parameters of the proposed machine in Fig. 10 should be given, while the required specifications should be applied to the optimal design. In my opinion, a revised version should be given for the required quality of the published paper in journal.

Reviewer 2 Report

Some general remarks are listed below. 

1. In section 2, various optimization techniques should be introduced and reviewed, such as the multi-level strategy, surrogate-assisted method, etc., rather than only RSM. Also, a brief trade-off analysis of these methods is recommended. 
2. The input contains five variables of different orders of magnitude. Probably the following calculation requires the variable normalization. 
2. For a better reading and understanding, the definition or formulation of the relative cross-validation error (in Fig. 3) should be provided. Also, the different sample portions should be given.
2. What is the meaning of your optimization in terms of changing the rotor radius (x1), as the stator dimensions are fixed?

Reviewer 3 Report

This paper proposes a new optimization algorithm of a rotor for synchronous reluctance machine. The topic is interesting, however there are several points which must be improved. The introduction part (sections 1 and 2) are quite weak and must be strengthened. The necessity of the research is not explained. The authors should consider existing approaches and explain, why they are not suitable for their purposes.

The quality of figures is inacceptable for journal publication. All plots and sections are recommended to be in a vector format. The raster pictures have to be in higher resolution.

It is hard to check the explanation, because parameter used for optimizations are not shown in the corresponding cross-section, thus the correctness of them may be checked only in the revised version of the paper.

1) You explained the necessity of motor optimization, however proposed only optimization of the rotor. Please explain, why the stator is not optimized.

2) You wrote: “The magnetic system is the main part of all electromagnetic devices. It consists of ferromagnetic material parts and air gaps.” This statement is incorrect, you forgot other types of machine construction. Furthermore, the construction depends on the type of machine.

3) The statement: “Motor torque, efficiency coefficient, power coefficient, and all main characteristics of a machine are defined by the difference in magnetic conductivity in axes d and q.” is incorrect. If you talk about motors in general, you have to consider other factors. Furthermore, you forgot about control issues, they significantly impact the performance criteria.

4) You wrote: “…large saliency ratio and large Ld-Lq”. Please specify the saliency ratio. Denote Ld and Lq. It is also recommended to use indexes for d and q.

5) You wrote that the inductance difference is important, thus it is not clear, why the saliency ratio is the purpose of optimization.

6) The sentence “Despite the use of high-performance computers (HPC) and even multi-cluster computing centers for parallel calculations, addressing numerous actual technical problems, particularly regarding optimization, requires too many computing resources.” Is not clear, please explain in more detail.

7) What is “meta-models”?

8) The sentence is not clear “Experiment planning helps in raising awareness about the subject”. Which subject are you talking about? Meta-models? How planning may help here?

9) You wrote: “Reporting articles have also been published on RSM: [12], [13], [14], [15] and [16], [17].” Which articles are reported? If those that listed at the end of sentence, the sentence should be corrected.

10) You provided several references [12]-[19] without consideration. What is the purpose of this list? You’d better briefly consider them.

11) In the sentence “General RSM application objectives:” a verb is missed.

12) You refer to RSM, but do not explain, what it is. You have to provide brief explanations.

13) It is not clear, if general RSM application objectives are connected or not?

14) Revise: “The object of investigation is synchronous reluctance motor with transversally laminated anisotropy, rotor-based”.

15) Improve quality of figures. All plots and sections are recommended to be in a vector format. The raster pictures have to be in higher resolution. For example, the details cannot be seen in the picture of stator.

16) You wrote: “Wb is the sum of the flux air barrier widths, and Wg…”, however the corresponding formula contains “wb” and “wg”.

17) You’d better show the corresponding parameters you are talking about (like “wb” and “wg” and other variables) in the cross section of the rotor.

18) Revise: “In it is reported…”.

Author Response

Please see the atttachment.

Round 2

Reviewer 1 Report

In the revised version, more modifications have been added, such as the designed characteristics of the investigated motor shown in Fig. 1 and the given parameters of the proposed machine in Fig. 10. However, some improvements can be given. On page 5, the use of w or ω should be consistency. On page 6, the explanations of equation (2) is confuse to me. The authors claims that the form of the true response function f is unknown and ε is an error term that represents the source of variability. Here, the use of the ε being an error seems to be a lack of rationale in the case. On page 7, the formats of the equations (3)-(6) should be common engineering symbols, not program symbols.

Author Response

Response is attached.

Reviewer 2 Report

According to the cover letter, the authors did not reply to the reviewer's comments in a straightforward way, but lightly wrote some introductory contents. The previous concerns remain pending. The paper still requires a revision. The concerns are reclaimed as follows. 

1. "In section 2, various optimization techniques should be introduced and reviewed, such as the multi-level strategy, surrogate-assisted method, etc., rather than only RSM. Also, a brief trade-off analysis of these methods is recommended." From the paper title, the novelty of the proposed RSM is not highlighted so that the content still belongs to the scope of optimization. Secondly, section 2 is named as 'Design Optimization' but it only reviews one method - RSM. Therefore, please complement a comprehensive review of various design optimization methods. 

2. "The input contains five variables of different orders of magnitude. Probably the following calculation requires the variable normalization." Please indicate this point in the paper for a better understanding. 

3. "For a better reading and understanding, the definition or formulation of the relative cross-validation error (in Fig. 3) should be provided. Also, the different sample portions should be given." I think you have misunderstood my concerns. Here, I mean you should provide the detailed steps and parameters of the relative cross-validation, rather than introduce it in a review tone. What you complemented in this version is textbook knowledge.

4. "What is the meaning of your optimization in terms of changing the rotor radius (x1), as the stator dimensions are fixed?" The author did not respond directly to this concern directly. Since the stator is already prototyped, the change of rotor radius (x1) will finally contribute to different air-gap lengths. It is definitely not a proper way. 

Author Response

Response is attached.

Reviewer 3 Report

1) Dear authors, when you respond to the comments you have to provide modified parts of the text as well. Furthermore, the modified parts in the paper have to be highlighted.

Please modify the paper, respect the time of reviewers.

2) The paper still contains raster pictures, which should be in vector formats. Please pay attention to document preparation and its export to pdf.

3) Despite my comments, you still did not provide definition of “metamodel”.

4) Provide more explanations on RSM. Does it cover construction of a model?

5) Pay attention to formatting. Ex. eq (3) is incorrectly formatted.

6) Despite you response, the design parameters were not written in the cross section (ex. wg and wb)

Round 3

Reviewer 2 Report

Some previous concerns remain pending. Also, the complemented contents have some problems. The paper still requires a major revision. 

1. For comment 1, the response is acceptable. 

2. "The input contains five variables of different orders of magnitude. Probably the following calculation requires the variable normalization." I cannot see any change indicating this point in the last version. 

3. The modification related to comment 3 is fine. 

4. The authors should justify and indicate in the context that the change of air-gap length is an appropriate way when implementing performance optimization, or that the range of air-gap change is limited to affect performance. 

Additional concerns are as follows. 

5. The authors state "In order to check this statement the saliency ratio range is chosen from 0.2 to 1.2 in Table 2." on page 5. Yet, I can only see the definition of insulation ratio rather than the saliency ratio. What is the saliency ratio and how is the saliency evaluated before obtaining the performance? 

6. Fig. 4-Fig. 6 illustrate the same thing which is the response relation among several objectives. To avoid repetition, I recommend the authors complement the relationship between objectives and input variables, like y1 versus x1. It is necessary and more straightforward to see the quality of optimization through the relation of input-output pairs. 

Author Response

Author's reply is attached.

Round 4

Reviewer 2 Report

Some previous concerns remain pending. The paper requires a minor revision. 

2. The response is ok and I understand the normalization process is done by the software automatically. 

6. Fig. 4-Fig. 6 illustrate the same thing which is the response relation among several objectives. To avoid repetition, I recommend the authors complement the relationship between objectives and input variables, like y1 versus x1. It is necessary and more straightforward to see the quality of optimization through the relation of input-output pairs. 
